# Does Aggressive Surgery Mean Worse Quality of Life and Functional Capacity in Retroperitoneal Sarcoma Patients?—A Retrospective Study of 161 Patients from China

**DOI:** 10.3390/cancers14205126

**Published:** 2022-10-19

**Authors:** Aobo Zhuang, Yuan Fang, Lijie Ma, Hua Yang, Weiqi Lu, Yuhong Zhou, Yong Zhang, Hanxing Tong

**Affiliations:** 1Department of General Surgery, Zhongshan Hospital, Fudan University, Shanghai 200032, China; 2Shanghai Public Health Clinical Center, Department of General Surgery, South Hospital of the Zhongshan Hospital, Fudan University, Shanghai 200093, China; 3Retroperitoneal Tumor Committee of Oncology Society of Chinese Medical Association, Xiamen University Research Center, Xiamen University, Xiamen 361005, China; 4Department of Gastrointestinal Surgery, Shanghai General Hospital Affiliated to Shanghai Jiaotong University, Shanghai 200940, China; 5Department of Medical Oncology, Zhongshan Hospital, Fudan University, Shanghai 200032, China

**Keywords:** retroperitoneal sarcoma, quality of life, multiple visceral resection

## Abstract

**Simple Summary:**

The mainstay of treatment for primary retroperitoneal sarcoma is surgery. In this study, we found that aggressive surgery did not imply a poorer quality of life and functional capacity compared to a surgical strategy of simple tumor resection. This has implications for patient consultation in daily practice, the clinical decision-making of surgeons, and even for subsequent prospective clinical studies on the quality of life.

**Abstract:**

The mainstay of treatment for primary retroperitoneal sarcoma (RPS) is surgery. However, whether multiple visceral resection (MVR) affects patients’ quality of life compared with simple tumor resection has not been reported. Patients with primary RPS who underwent radical resection between 2009 and 2021 were included. Patients who were alive at the last follow-up were asked to complete the European Organization for the Research and Treatment of Cancer Core Quality of Life Questionnaire (EORTC QLQ-C30). The primary endpoint of the study was the global health (GH) score. A total of 161 patients were included, including 77 in the MVR group and 84 in the non-MVR group. When comparing EORTC scores on functional domains and symptom scales between MVR and non-MVR groups, only constipation scores differed (*p* = 0.011). Comparing GH scores within 6 months after surgery between the two groups, GH was better in non-MVR patients (58.3 vs. 76.4, *p* = 0.082). However, patients with longer postoperative intervals in the MVR group had higher scores (*p* < 0.001), and patients with postoperative intervals of more than one year scored similar to those in the non-MVR group (64.7 vs. 59.2, *p* = 0.522). As the postoperative interval increased, there was an improvement in all indicators in MVR patients, while there was no significant improvement in non-MVR patients. Aggressive surgical approaches impair quality of life within 6 months postoperatively, but the long-term quality of life is similar to that of patients undergoing simple tumor resection. This should be factored into RPS treatment decisions.

## 1. Background

Retroperitoneal sarcoma (RPS) has been reported as a rare malignancy with 0.76 new cases per 100,000 people per year [1] and is usually massive at the time of diagnosis and invades multiple adjacent organs. Surgery is still the main treatment for primary localized RPS.

In 2009, retrospective studies from two major European reference centers showed that a surgical approach combined with the resection of uninvolved adjacent organs in RPS was correlated with improved local control [2,3]. Subsequent studies have confirmed the perioperative safety of this aggressive surgical strategy [4]. Multivisceral resection (MVR), which requires resection of the tumor with adherent structures, has become a cornerstone of RPS management. Some studies have reported that the median number of organ resections may be as high as 4–5 for high-volume sarcoma centers that advocate for combined multi-organ resection [5]. As studies have progressively improved, we have gained a comprehensive understanding of the objective outcomes (e.g., recurrence and survival) of MVR surgery, but subjective patient-reported outcomes (PROs) including health-related quality of life (HRQoL) are also matters of concern.

PROs are defined as “any report of the status of a patient’s health condition that comes directly from the patient, without interpretation of the patient’s response by a clinician or anyone else” [6] and are used to evaluate treatment efficacy as well. It includes a range of outcomes such as symptoms and functioning. HRQoL, the most widely used PRO, is a multidimensional concept that includes patients’ perceptions of the impact of their disease and the treatment on physical, psychological, and social functioning [7]. Incorporating PROs into clinical practice can facilitate communication, improve symptom control and patient satisfaction, and reduce hospitalizations.

High-quality HRQoL data of patients with sarcoma are sparse, and there are even fewer studies on RPS. There are only three reports on HRQoL for surgically resected RPS and the sample is limited [8,9,10]. This means we do not know whether aggressive surgery results in a worse postoperative quality of life compared to simple tumor resection or not.

We aimed to take advantage of a high-volume sarcoma center to explore changes in the quality of life of patients with MVR compared with simple tumor resection after surgery and provide a reference for further prospective clinical trials and a certain basis for clinical decision-making.

## 2. Methods

### 2.1. Subject

This was a retrospective study including all patients with RPS who underwent curative surgery in the Zhongshan Hospital, Fudan University, Shanghai, China from August 2009 to December 2021. Inclusion criteria were as follows: (1) Primary disease, (2) histologically confirmed sarcoma, (3) tumors originating in the retroperitoneum, (4) the absence of synchronous malignancies, and (5) complete clinical pathological information and follow-up information. Besides, we excluded patients with Ewing sarcoma, epithelioid sarcoma, ligamentoid fibroma, gynecologic sarcoma, and gastrointestinal stromal tumors. A total of 319 patients underwent curative surgery but 99 passed away (Appendix A) during this period. Of the 220 patients who were still alive at the time of study commencement, 29 were lost to follow-up, 21 declined to participate in this study, and 9 were unable to complete questionnaires. The baseline characteristics of responders and non-responders are shown in Appendix A. Compared with responders, non-responders had a higher proportion of males (*p* = 0.013) and more symptomatic patients at presentation (0.031). Because of the COVID-19 pandemic, most patients would not be available for postoperative outpatient follow-up in the near future. Therefore, this study obtained informed consent from the patients through a telephone follow-up from March to April 2022 and then sent a link to encourage patients to complete the quality-of-life scale online. For patients with difficulty in Internet access or reading, we completed the HRQoL scale through telephone inquiries. This study was approved by the Ethics Committee of South Hospital of Zhongshan Hospital and was conducted in accordance with the Declaration of Helsinki.

MVR was defined as tumor resection in combination with at least two organs [2]. The European Organization for the Research and Treatment of Cancer Core Quality of Life Questionnaire (EORTC QLQ-C30) was used to assess patients’ HRQoL postoperatively. The questionnaire comprises five functional scales, three symptom scales, six single-symptom items, and a global health-related QoL score [11]. For functional and global health (GH) scores, higher scores mean better function and GH. The GH score is one of the core evaluation indicators of the EORTC QLQ-C30 scale, which is mainly based on the patient’s self-assessment of the health status and quality of life in the last week. However, the higher the symptom score, the more pronounced the symptom. Complications that arose post-operatively were classified according to the Clavien–Dindo Classification [12]. According to Hui et al. [9], the Qol of RPS patients improved significantly 2 years after the operation, so we divided the patients into 6 groups according to the time interval from the operation to filling out the questionnaire; specifically, T1, 0–6 months; T2, 6–12 months; T3, 12–18 months; T4, 18–36 months; T5, 2–5 years; and T6, more than 5 years. The preoperative status of patients was assessed by the American Society of Anesthesiologists Physical Status (ASA score).

### 2.2. Statistical Methods

The primary endpoint of the study was the GH score. Disease-free survival was calculated from the date of surgery to the date of the last follow-up or disease relapse or death by any cause. Disease-free survival was calculated using Kaplan–Meier and compared by log-rank tests. For continuous variables, if they conformed to the normal distribution, they were described by the mean and standard deviation and compared using Student’s *t*-test. Since EORTC QLQ-C30 scores did not conform to a normal distribution, they were summarized using the mean, standard deviation, median, and interquartile range and non-parametric tests were used. Categorical variables were summarized by the number and percentage of patients in each category and compared by Fisher’s exact test or Pearson Chi-square. In addition, the linear regression models were used to evaluate differences in scores across time.

All tests were two-tailed, and *p* < 0.05 was considered statistically significant.

## 3. Results

### 3.1. Baseline Characteristics

A total of 161 patients were included in this study, including 77 MVR patients and 84 non-MVR patients. The median time to complete the questionnaire for all patients was 26.2 (range, 4.0–131.7) months after surgery. For baseline characteristics (Table 1), there was no significant difference in gender distribution (*p* = 0.141), ASA score (*p* = 0.210) and mean age (55.9 vs. 54.5 years, *p* = 0.494) between the two groups.

There was also no difference in the FNCLCC grade (*p* = 0.216) or proportion of radiotherapy and chemotherapy. However, patients in the MVR group had a greater tumor burden (21.0 vs. 11.9 cm, *p* < 0.001), higher rates of WDLPS (55.8% vs. 34.5%) and DDLPS (33.8% vs. 7.1%), and lower rates of LMS (5.2% vs. 22.6%). In terms of surgical characteristics, as MVR itself is a more aggressive surgical strategy, there was a higher median number of resected organs (3 vs. 1, *p* < 0.001), longer operation time (4.4 vs. 2.7 h, *p* < 0.001), and more estimated blood loss (780.6 vs. 424.6 mL) and packed RBC transfusions (32% vs. 15) compared to non-MVR patients. More MVR patients were transferred to the ICU after surgery (74% vs. 33%, *p* < 0.001), and had a longer postoperative hospital stay compared to non-MVR patients (19 vs. 13 days, *p* = 0.011).

In terms of oncological outcomes, the 5-year disease-free survival rates of MVR and non-MVR patients were 70.1% (95%CI, 54.5–85.7) and 76.7% (95%CI, 64.7–88.7), respectively. There was no statistical difference between the two groups (*p* = 0.211) (Figure 1).

### 3.2. Summary of GH Scores

The mean GH score of all patients was 71.0 (SD = 19.0). The differences in GH scores between various clinicopathological factors are shown in Table 2.

Changes greater than 10 points were considered significant for all EORTC functional domains and symptom scales [13]. Overall, there were no differences in patients’ gender, age, ASA grade, and histologic subtypes. However, there was a statistically significant difference between the presence or absence of serious complications (*p* = 0.008), recurrence (*p* = 0.005), and multifocal disease (*p* = 0.001). The difference in GH scores of the above three factors is clinically relevant (the difference was greater than 10 points).

### 3.3. Comparison of EORTC Scores between MVR and Non-MVR Groups

Because surgery is the treatment of choice for locally recurrent RPS and reoperation affects the determination of postoperative time, we excluded 31 patients who had recurrence after surgery in the analysis of MVR versus non-MVR groups.

#### 3.3.1. Across Functional Domains and Symptom Scales

We compared differences in functional domains and symptom scales between MVR patients and non-MVR patients (Table 3).

The mean GH scores of the MVR and non-MVR groups were 75.9 (SD, 16.0) and 70.8 (SD, 19.5), respectively, with no statistical difference between the two groups (*p* = 0.213). In addition, there were no differences in the scores of the five functional domains between the two groups. In terms of symptoms, patients in the non-MVR group had higher constipation scores (29.5 vs. 14.0, *p* = 0.011), while there was a decreasing trend in diarrhea (15.9 vs. 20.4, *p* = 0.214).

#### 3.3.2. Across Time

The number of patients and EORTC scores for each group are shown in Table 4 and Figure 2, which shows the trend of the mean time trends of EORTC scores in the MVR and non-MVR groups.

Overall, the non-MVR group had higher functional scores and symptom scores within 6 months after surgery. With the prolonged postoperative interval, the indicators improved in MVR patients, while there was no significant improvement in non-MVR patients. For those patients with a surgery interval greater than 5 years, the vast majority of EORTC QLQ-C30 scores in the MVR group were better or no worse than those in the non-MVR group.

In terms of the GH score, the MVR group was lower than the non-MVR group within 6 months after surgery (58.3 vs. 76.4, *p* = 0.082). The scores improved in the MVR group with a longer postoperative interval (*p* < 0.001) and were similar to the non-MVR group one year after surgery (64.7 vs. 59.2). For intervals more than five years, the MVR group had a significantly higher GH score than the non-MVR group (91.7 vs. 71.5, *p* = 0.036).

Similarly, physical function scores were lower in the MVR group than in the non-MVR group at 6 months postoperatively (69.3 vs. 84.4) but improved significantly from 6 to 12 months after surgery. The physical functioning score improved as the surgical interval increased in both the MVR group and the non-MVR group and were similar 1 year after surgery.

For role functioning, emotional functioning, cognitive functioning, and social functioning, the scores of the MVR group were slightly lower than those of the non-MVR group within 6 months after the operation, followed by a decreasing trend in the non-MVR group, with no significant difference between the two groups in patients with a surgical interval greater than five years.

In the symptom scores of fatigue, pain, appetite loss, diarrhea, and financial difficulties, the postoperative scores of patients in the MVR group were lower than those in the non-MVR group. However, it was significantly higher (*p* < 0.05) in the MVR group with longer surgical intervals, while the trend of change was not significant in the non-MVR group. Therefore, compared with the non-MVR group, the scores of patients in the MVR group were higher within 1 year after surgery, but there was no significant difference between the two groups after more than 1 year.

The only symptoms with higher scores in the non-MVR group with an interval between surgery of less than 6 months were dyspnea and constipation. For dyspnea, the difference between the two groups was not significant one year after surgery, but constipation scores remained higher in the non-MVR group than in the MVR group five years after surgery.

The scores of nausea, vomiting, and insomnia were lower in the MVR group with a short postoperative period; however, as the follow-up interval increased, there was no significant difference between the two groups.

## 4. Discussion

Surgery remains the only effective treatment for primary RPS, and the best chance of cure is at the time of primary presentation [14]. Many high-volume sarcoma centers are actively pursuing MVR procedures. However, surgeons must make trade-offs between oncological outcomes and functional preservation when resecting certain vital retroperitoneal organs. Therefore, a better understanding of HRQoL throughout the course of treatment can help guide collaborative decision-making with patients. This study was the first to focus on the quality-of-life differences between MVR and non-MVR patients with primary RPS. Although it was a retrospective study, selection bias cannot be avoided, and we can still draw some insights from the findings. First, although the quality of life of MVR patients was significantly lower than non-MVR patients within 6 months after surgery, the vast majority of MVR patients had better functional and symptom scores 1 year after surgery than non-MVR patients. Therefore, for patients with an expected survival time of more than one year, the choice of surgical strategy for the best oncological outcome is based on individual circumstances and pathological types. For these patients, there is no need to worry about the impact of MVR on their long-term quality of life after surgery. Second, even more than 3 years after surgery, the quality of life continued to improve in the vast majority of MVR patients, but the trend of postoperative quality of life improvement was not obvious in the non-MVR group.

There are only three studies on the quality of life after RPS. The first is a study conducted in 2017 by Philip et al., which included a cohort of 48 patients, and the purpose of the study was to investigate the relationship between neoadjuvant radiotherapy toxicity and quality of life in patients after resection of RPS. In Philip’s study, the average global health score of patients with postoperative recurrence-free survival greater than 36 months was 75.0, similar to the cohort of our study (78.1). The median number of organ resections in Philip’s cohort was 4, but the sample size was small and differences between the MVR and non-MVR groups have not been explored [8]. The second study was also a retrospective study from a single center in Asia. It collected the EORTC-QLQ-C30 questionnaires from patients with primary RPS who were treated at the National Cancer Centre Singapore from January 1999 to August 2018 and found that the quality of life of elderly patients and male patients was worse after surgery and that the quality of life was better in patients more than 2 years postoperatively than in patients within 2 years. The average Global health score of all patients was 79.9 (SD, 18.6), which was slightly higher than that of this cohort (71.0, SD, 19.0), but did not reach clinical significance (difference less than 10). However, this Asian single-center cohort included only 32 patients, and the median number of organ resections was only 1 [9]. The most recent study was reported by Gronchi et al. in November 2020. The purpose of Gronchi’s study was to explore the changes in the quality of life of MVR patients after surgery. A total of 58 patients were included in the study, and the quality-of-life scales of these patients were prospectively collected one day before surgery, 4 months after surgery, and 1 year after surgery. They also found that the quality of life of MVR patients continued to improve after surgery. However, this study was a single-arm study with a follow-up period of only one year [10].

We believe that the reasons for the enhanced postoperative quality of life in MVR patients compared with non-MVR patients are as follows: (1) Compared with the non-MVR group, patients in the MVR group have more combined organ resection, longer operation time, more bleeding, and more postoperative complications, which means that the quality of life of the patients in the MVR group declined more than the non-MVR patients in the short term, which also means more room for improvement. (2) From the patient’s point of view, the most direct strategy for oncologic surgery is to remove the tumor, while patients who have adopted more aggressive surgical strategies may have a better understanding of their disease and treatment options. Patients gaining a clear understanding of their treatment plan has been reported to have a positive impact on their lives. (3) The last reason is changes in values, as MVR patients perceive that their tumor has been completely removed and have hope for long-term recurrence-free survival, which was mentioned in a study performed by Duckworth et al., in 2012. They investigated 102 patients at least a year or more after their surgeries. Their results showed that while the physical symptoms remained and physical scores were below normal, mental health scores improved. This suggests that while physical and functional deficits remain, survivors are able to adjust, adapt, and achieve emotional well-being. This is referred to as a response shift, whereby the patients accommodate new norms and experience shifts in internal standards [15].

Our study has several limitations. First, retrospective studies inevitably suffered from selection bias, for example, the quality of life of the 99 patients who died in this cohort could not be obtained. Second, although the purpose of this study was to compare the differences in postoperative quality of life between MVR and non-MVR patients, we did not include patients’ baseline conditions (preoperative quality of life). Therefore, it was not feasible to compare longitudinally the changes in the quality of life before and after surgery between the two groups of patients. Third, this study included only the EORTC QLQ-C30 quality of life scale; however, only using the QLQC-30 questionnaire may not be sufficient to capture QoL changes in sarcoma patients.

## 5. Conclusions

In conclusion, as a retrospective study, we are well aware that it has many shortcomings. However, as RPS is a rare tumor, the reported postoperative quality of life in 161 patients has been considerable. In this exploratory study, we found that aggressive surgery did not imply a poorer quality of life and functional capacity compared to a surgical strategy of simple tumor resection. This has implications for patient consultation in daily practice, the clinical decision-making of surgeons, and even for subsequent prospective clinical studies on the quality of life.

## Figures and Tables

**Figure 1 cancers-14-05126-f001:**
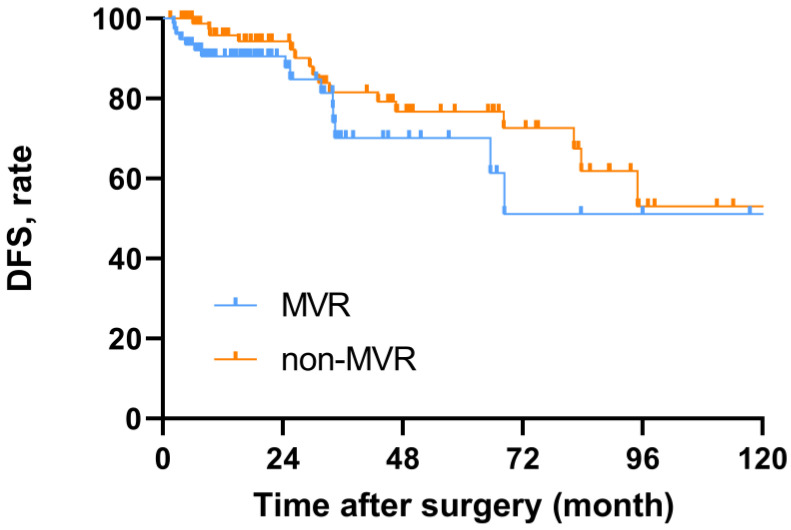
Disease-free survival for MVR and non-MVR patients.

**Figure 2 cancers-14-05126-f002:**
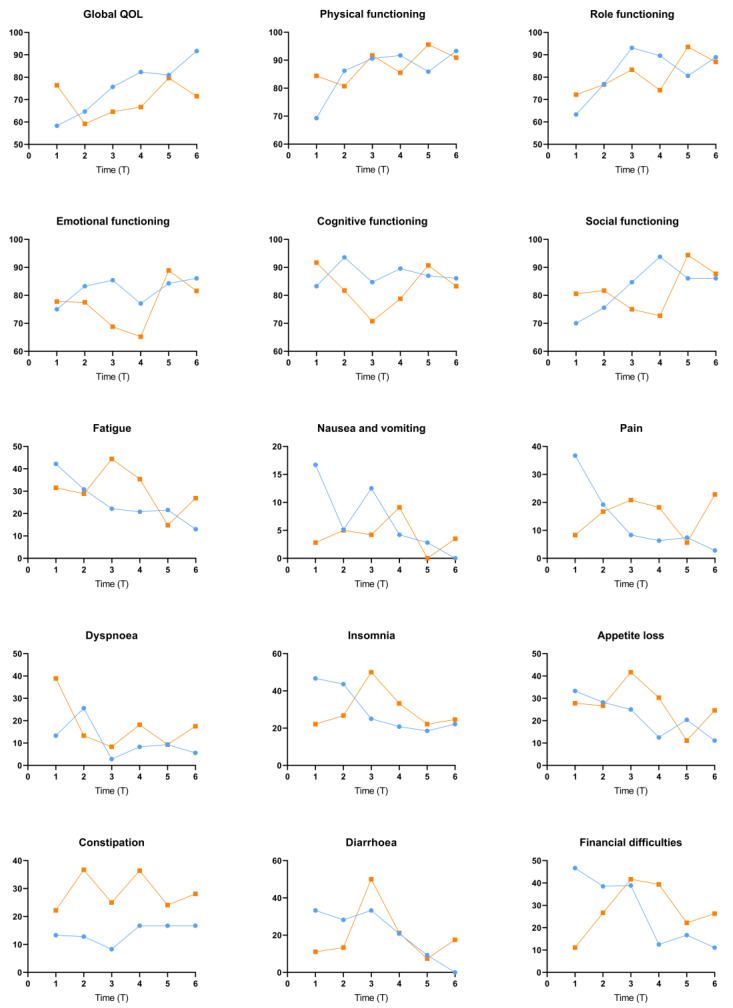
Mean EORTC scores for MVR and non-MVR patients over time.

**Table 1 cancers-14-05126-t001:** Baseline characteristics for MVR and non-MVR patients.

Characteristics	MVR (*n* = 77)	Non-MVR (*n* = 84)	*p*
Gender			0.141
Male	41 (53.2)	35 (41.7)	
Female	36 (46.8)	49 (58.3)	
Age, years mean (SD)	55.9 (±12.3)	54.5 (±14.1)	0.494
ASA score			0.210
1	54 (70.1)	51 (60.7)	
>1	23 (29.9)	33 (39.3)	
Symptoms			0.637
Yes	23 (29.9)	28 (33.3)	
No	54 (70.1)	56 (66.7)	
Tumor burden, cm mean (SD)	21.0 (±8.9)	11.9 (±7.1)	<0.001
Histologic subtypes			<0.001
WDLPS	43 (55.8)	29 (34.5)	
DDLPS	26 (33.8)	6 (7.1)	
LMS	4 (5.2)	19 (22.6)	
SFT	1 (5.6)	17 (20.2)	
Others	3 (3.9)	13 (15.5)	
FNCLCC			0.216
Grade 1	29 (37.7)	40 (47.6)	
Grade 2	29 (37.7)	22 (26.2)	
Grade 3	12 (15.6)	18 (21.4)	
Unknow	7 (9.1)	4 (4.8)	
Location			0.681
Left	41 (53.2)	42 (50.0)	
Right	36 (46.8)	42 (50.0)	
Multifocality			0.644
Yes	6 (7.8)	5 (6.0)	
No	71 (92.2)	79 (94.0)	
Radiation			0.162
Yes	5 (6.5)	11 (13.1)	
No	72 (93.5)	73 (86.9)	
Chemotherapy			0.655
Yes	9 (11.7)	8 (9.5)	
No	68 (88.3)	76 (90.5)	
Operation			0.370
Laparoscopic surgery	1 (1.3)	4 (4.8)	
Open surgery	76 (98.7)	80 (95.2)	
Complete resection			1.000
Yes	83 (98.7)	83 (98.8)	
No	1 (1.3)	1 (1.2)	
Major vascular surgery			0.077
Yes	11 (14.3)	5 (6.0)	
No	66 (85.7)	79 (94.0)	
Pancreaticoduodenectomy			0.068
Yes	3 (3.9)	0 (0.0)	
No	74 (96.1)	84 (100.0)	
Number of combined resections median, (IQR)	3 (2–4)	0 (0–1)	<0.001
Resected organs			
Colon	58 (75.3)	15 (17.9)	<0.001
Kidney	60 (77.9)	9 (10.7)	<0.001
Adrenal gland	36 (46.8)	1 (1.2)	<0.001
Spleen	16 (20.8)	0 (0)	<0.001
Pancreas	16 (20.8)	0 (0)	<0.001
Small intestine	13 (16.9)	2 (2.4)	0.002
Diaphragm	6 (7.8)	2 (2.4)	0.154
Abdominal Wall	3 (3.9)	1 (1.2)	0.350
Operative time, hours mean (SD)	4.4(±1.7)	2.7 (±1.1)	<0.001
Estimated blood loss, ml median, (IQR)	780.6 (±713.3)	424.6 (±780.5)	0.004
Packed RBC transfusion			0.011
Yes	25 (32.5)	13 (15.5)	
No	52 (67.5)	71 (84.5)	
ICU Stay			<0.001
Yes	57 (74.0)	28 (33.3)	
No	20 (26.0)	56 (66.7)	
Severe postoperative adverse events			0.088
Yes	7 (9.1)	2 (2.4)	
No	70 (90.9)	82 (97.6)	
Postoperative Hospital Stay, days mean (SD)	19.1 (±11.2)	13.5 (±9.1)	0.002
Disease recurrence			0.792
Yes	15 (19.5)	16 (19.0)	
No	62 (80.5)	68 (81.0)	

**Table 2 cancers-14-05126-t002:** Global health score for primary retroperitoneal sarcoma patients.

Characteristics	Mean	Standard Deviation	Median	IQR	*p*
Gender					0.672
Male	70.7	17.2	70.8	58.3–83.3	
Female	71.2	20.6	75.0	58.3–83.3	
Age, years median					0.101
≤60	72.8	19.4	75.0	58.3–83.3	
>60	67.9	18.0	66.7	58.3–83.3	
ASA score					0.088
1	73.2	18.1	75.0	58.3–83.3	
>1	66.7	20.2	66.7	50.0–83.3	
Symptoms					0.119
Yes	67.3	20.9	66.7	50.0–83.3	
No	72.7	17.9	75.0	58.3–83.3	
Tumor burden					0.243
0–10	73.1	19.9	75.0	58.3–83.3	
>10	69.8	18.4	66.7	58.3–83.3	
Histologic subtypes					0.963
WDLPS	71.6	17.8	75.0	58.3–83.3	
DDLPS	70.1	18.7	75.0	58.3–83.3	
LMS	68.8	24.0	66.7	50.0–83.3	
SFT	69.4	18.3	66.7	50.0–83.3	
Others	74.5	19.6	70.8	58.3–100.0	
FNCLCC					0.550
Grade 1	71.5	19.5	75.0	58.3–83.3	
Grade 2	71.6	20.4	75.0	50.0–83.3	
Grade 3	71.1	18.3	75.0	56.3–83.3	
Unknow	65.2	10.4	66.7	58.3–66.7	
Location					0.510
Left	72.5	17.4	75.0	58.3–83.3	
Right	69.4	20.5	66.7	58.3–83.3	
Multifocality					0.001
Yes	51.5	20.7	50.0	50.0–66.7	
No	72.4	18.1	75.0	58.3–83.3	
Radiation					0.165
Yes	76.6	17.0	83.3	66.7–89.6	
No	70.4	19.2	66.7	58.3–83.3	
Chemotherapy					0.841
Yes	72.5	13.4	66.7	66.7–83.3	
No	70.8	19.6	75.0	58.3–83.3	
Severe postoperative adverse events					
Yes	54.6	16.2	58.3	37.5–70.8	0.008
No	72.0	18.8	75.0	58.3–83.3	
Recurrence					
Yes	61.4	20.4	66.7	50.0–75.0	0.005
No	73.2	18.1	75.0	58.3–83.3	

**Table 3 cancers-14-05126-t003:** EORTC scores for MVR and non-MVR patients.

Characteristics	MVR (*n* = 62)				non-MVR (*n* = 69)				*p*
	Mean	SD	Median	IQR	Mean	SD	Median	IQR	
Global health	75.9	16.0	75.0	64.6–83.3	70.8	19.5	75.0	52.1–83.3	0.213
Functional Scales									
Physical functioning	87.0	14.4	90.0	80.0–100.0	89.4	12.9	93.3	80.0–100.0	0.282
Role functioning	82.8	18.8	83.3	66.7–100.0	83.3	22.7	100.0	66.7–100.0	0.424
Emotional functioning	82.8	16.3	87.5	72.95–93.8	79.2	20.0	83.3	66.7–91.7	0.360
Cognitive functioning	87.9	14.5	100.0	83.3–100.0	84.3	15.6	83.3	66.7–100.0	0.179
Social functioning	83.3	20.9	83.3	66.7–100.0	84.5	20.7	91.7	83.3–100.0	0.549
Symptom Scales									
Fatigue	24.4	20.9	22.2	11.1–33.3	27.1	22.6	33.3	11.1–33.3	0.495
Nausea and vomiting	6.2	16.6	0	0–0	3.6	8.5	0.0	0.0–0.0	0.831
Pain	11.8	20.1	0	0–16.7	15.2	18.9	16.7	0.0–16.7	0.149
Dyspnoea	11.3	19.0	0	0–33.3	16.4	21.1	0.0	0–33.3	0.150
Insomnia	28.0	29.7	33.3	0–33.3	27.1	32.5	33.3	0–33.3	0.597
Appetite loss	22.0	24.1	33.3	0–33.3	23.7	24.3	33.3	0–33.3	0.721
Constipation	14.0	22.2	0	0–33.3	29.5	34.1	33.3	0–58.3	0.011
Diarrhoea	20.4	26.6	0	0–33.3	15.9	26.0	0.0	0–33.3	0.214
Financial difficulties	26.9	33.5	0	0–33.3	28.0	33.6	0.0	0–58.3	0.955

**Table 4 cancers-14-05126-t004:** Mean EORTC scores for MVR and non-MVR patients across time.

	Number		Global QOL			Physical Functioning			Role Functioning			Emotional Functioning			Cognitive Functioning			Social Functioning			Fatigue			Nausea and Vomiting			Pain			Dyspnoea			Insomnia			Appetite Loss			Constipation			Diarrhoea		
Time	MVR	Non MVR	MVR	Non MVR	*p* *	MVR	Non MVR	*p* *	MVR	Non MVR	*p* *	MVR	Non MVR	*p* *	MVR	Non MVR	*p* *	MVR	Non MVR	*p* *	MVR	Non MVR	*p* *	MVR	Non MVR	*p* *	MVR	Non MVR	*p* *	MVR	Non MVR	*p* *	MVR	Non MVR	*p* *	MVR	Non MVR	*p* *	MVR	Non MVR	*p* *	MVR	Non MVR	*p* *
1	5	6	58.3	76.4	0.082	69.3	84.4	0.177	63.3	72.2	0.662	75.0	77.8	1.000	83.3	91.7	0.792	70.0	80.6	1.000	42.2	31.5	0.329	16.7	2.8	0.429	36.7	8.3	0.329	13.3	38.9	0.126	46.7	22.2	0.126	33.3	27.8	0.792	13.3	22.2	0.792	33.3	11.1	0.662
2	13	10	64.7	59.2	0.522	86.2	80.7	0.522	76.9	76.7	1.000	83.3	77.5	0.257	93.6	81.7	0.131	75.6	81.7	0.738	30.8	28.9	0.693	5.1	5.0	0.879	19.2	16.7	0.738	25.6	13.3	0.284	43.6	26.7	0.284	28.2	26.7	0.927	12.8	36.7	0.166	28.2	13.3	0.232
3	12	4	75.7	64.6	0.170	90.6	91.7	1.000	93.1	83.3	0.212	85.4	68.8	0.078	84.7	70.8	0.212	84.7	75.0	0.684	22.2	44.4	0.170	12.5	4.2	1.000	8.3	20.8	0.078	2.8	8.3	0.684	25.0	50.0	0.684	25.0	41.7	0.262	8.3	25.0	0.379	33.3	50.0	0.521
4	8	11	82.3	66.7	0.129	91.7	85.5	0.600	89.6	74.2	0.717	77.1	65.2	0.492	89.6	78.8	0.238	93.8	72.7	0.075	20.8	35.4	0.657	4.2	9.1	0.442	6.3	18.2	0.238	8.3	18.2	0.310	20.8	33.3	0.778	12.5	30.3	0.545	16.7	36.4	0.272	20.8	21.2	0.840
5	18	18	81.0	79.6	0.938	85.9	95.6	0.134	80.6	93.5	0.047	84.3	88.9	0.239	87.0	90.7	0.606	86.1	94.4	0.074	21.6	14.8	0.239	2.8	0.0	0.584	7.4	5.6	0.938	9.3	9.3	0.839	18.5	22.2	0.719	20.4	11.1	0.406	16.7	24.1	0.424	9.3	7.4	0.791
6	6	19	91.7	71.5	0.036	93.3	90.9	0.780	88.9	86.8	0.877	86.1	81.6	0.555	86.1	83.3	0.780	86.1	87.7	0.555	13.0	26.9	0.106	0.0	3.5	0.598	2.8	22.8	0.176	5.6	17.5	0.437	22.2	24.6	0.926	11.1	24.6	0.246	16.7	28.1	0.733	0.0	17.5	0.274
*p* #	-	-	<0.001	0.278	-	0.070	0.048	-	0.101	0.071	-	0.340	0.070	-	0.861	0.674	-	0.222	0.083	-	0.015	0.338	-	0.207	0.226	-	0.018	0.672	-	0.100	0.405	-	0.107	0.476	-	0.039	0.404	-	0.441	0.691	-	0.004	0.488	-

* A non-parametric test was used to compare the difference of a certain score between the MVR and non-MVR groups at a certain time point. # Linear regression was used to compare trends in a score over time in patients with MVR or non-MVR.

## Data Availability

The datasets used and analyzed during the current study are available from the corresponding author upon reasonable request.

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
