# Peer review of "Does Aggressive Surgery Mean Worse Quality of Life and Functional Capacity in Retroperitoneal Sarcoma Patients?—A Retrospective Study of 161 Patients from China"

_cancers, 2022, doi:10.3390/cancers14205126_

Round 1
Reviewer 1 Report
Please correct typo in Table 1 and in the appendix table, in the chemotherapy Yse- Yes.
The details of GH should be opened somewhere, as it is a main end point.
ASA Score is not familiar to all readers, please open the term.
The text om the paragraph 3.3 does not fit to it's heading. Please change heading or text.
L 149 typograph: ->surgery.
In the paragraph 3.5 the text in the beginning should be in the Methods section.
As this is a one-time questionnaire the wording of the results is not correct. Authors do not have a real longitudional data, but different patients in the different postoperative phase. Also you can not be sure what was the previous score in that specific patient. So the wording should be changed throughtout the paper.
Do the authors have any information about the deceased? If any, it would be easier to assess the reliability of the results.
Author Response
First of all, I have to express my thanks to reviewer 1 for your time, your comments are very helpful to me.
1.Please correct typo in Table 1 and in the appendix table, in the chemotherapy Yse- Yes.
Thanks to the reviewer's reminder, the above two typos have been corrected.
2.The details of GH should be opened somewhere, as it is a main end point.
Thanks for your opinion. Indeed, as the primary endpoint of the study, GH really needs to be explained. I have added the following in the Methods section of the manuscript: The GH score is one of the core evaluation indicators of the EORTC QLQ-C30 scale, which is mainly based on the patient's self-assessment of the health status and quality of life in the last week.
3.ASA Score is not familiar to all readers, please open the term.
Thanks again for the reminder. I have added the following description to page 3 of the manuscript, line 108: The preoperative status of patients was assessed by the American Society of Anesthesiologists Physical Status (ASA score).
4.The text om the paragraph 3.3 does not fit to it's heading. Please change heading or text.
Thanks to reviewer 1 for the reminder. The title is wrong because of a typography issue. The title of 3.3 is correct, it is the mistakes of 3.4 and 3.5 that caused the misunderstanding. I have changed 3.4 and 3.5 to 3.31 and 3.32 respectively.
5.L 149 typograph: ->surgery.
Thanks again for your comments, the spelling above has been revised.
6.In the paragraph 3.5 the text in the beginning should be in the Methods section.
As you said, it's more appropriate to put the above part in the methods section, I've modified it accordingly.
7. As this is a one-time questionnaire the wording of the results is not correct. Authors do not have a real longitudional data, but different patients in the different postoperative phase. Also you can not be sure what was the previous score in that specific patient. So the wording should be changed throughout the paper.
Thank you very much for your opinion. Indeed, this study is more like a cross-sectional study than a longitudinal study. Some descriptions used in the original manuscript, such as "gradually", are easy to mislead readers. I have revised the corresponding presentations in the Abstract and Results sections.
8. Do the authors have any information about the deceased? If any, it would be easier to assess the reliability of the results.
Thank you for your reminder. I have organized the clinicopathological information of the deceased patient and uploaded it in the attachment (supplementary Table 1).
Reviewer 2 Report
The article titled “Does Aggressive Surgery Means Worse Quality of Life and Functional Capacity in Retroperitoneal Sarcoma Patients? - A Retrospective Study of 161 Patients from China" by Zhuang et. al. is very interesting.
The paper fits the scope of the Cancers journal.
In this original work, the authors check whether multiple visceral resection (MVR) affects patients’ quality of life. The authors confirmed that an aggressive surgical approach worsens the quality of life within six months after surgery, but the long-term quality of life is similar to that of patients undergoing simple tumor resection.
There are some comments for this interesting manuscript:
>The study is correctly designed.
>The results provide an advancement of the current knowledge.
>It is advisable to end the introduction with the objective of the study.
>The cited references are mostly relevant recent publications.
The fact that this was a retrospective study made it impossible to compare preoperative quality of life between patients with and without MVR. It was not possible to longitudinally compare changes in quality of life before and after surgery between the two groups of patients, as the authors rightly mentioned in the study’s limitations.
I suggest this paper to be published in Cancers after suitable revision.
Author Response
First of all, I would like to thank reviewer 2 for their encouragement and support for this research. Your comments are very useful to me.
1.The article titled “Does Aggressive Surgery Means Worse Quality of Life and Functional Capacity in Retroperitoneal Sarcoma Patients? - A Retrospective Study of 161 Patients from China" by Zhuang et. al. is very interesting. The paper fits the scope of the Cancers journal. In this original work, the authors check whether multiple visceral resection (MVR) affects patients’ quality of life. The authors confirmed that an aggressive surgical approach worsens the quality of life within six months after surgery, but the long-term quality of life is similar to that of patients undergoing simple tumor resection.
Thank you for your time in this manuscript and for acknowledging this research.
2.The study is correctly designed.
Thank you again for your endorsement of this research.
3.The results provide an advancement of the current knowledge.
We thank reviewer 2 for their affirmation of this study.
4.It is advisable to end the introduction with the objective of the study.
Thanks for your comments, I have revised the last paragraph of the introduction as follows: We aimed to take advantage of a high-volume sarcoma center to explore the changes in the quality of life of patients with MVR compared with simple tumor resection after surgery and to provide a reference for further prospective clinical trials and a certain decision basis for clinical decision-making.
5.The cited references are mostly relevant recent publications.
We thank the reviewers for their recognition of this study.
6.The fact that this was a retrospective study made it impossible to compare preoperative quality of life between patients with and without MVR. It was not possible to longitudinally compare changes in quality of life before and after surgery between the two groups of patients, as the authors rightly mentioned in the study’s limitations.
Thanks to reviewer 2 for their comments. As a retrospective study, this study inevitably has selection bias, but as an exploratory study, this study can provide a reference for subsequent prospective clinical studies.
Round 2
Reviewer 1 Report
Now the paper is acceptable, but somer minor typos are still to corrected:
Page 10, line 19 and and page 7, line 166.
Page 10, line 30 a word is missing? Same in page 11, line 86?
Author Response
Thank you again for your two rounds of review of this manuscript. Page 10, line 19, 'was' changed to' were '; In page 7, line 166, 'were75.9 (SD, 16.0)' is changed to 'were75.9 (SD, 16.0)'; page 10, line 30 and page 11, line 86 added 'non-MVR'.